# A self-healing catalyst for electrocatalytic and photoelectrochemical oxygen evolution in highly alkaline conditions

Chao Feng [1], Faze Wang[1,2], Zhi Liu[1], Mamiko Nakabayashi [3], Yequan Xiao[1], Qiugui Zeng[1], Jie Fu[1], Qianbao Wu [1], Chunhua Cui [1], Yifan Han[4], Naoya Shibata [3], Kazunari Domen [5,6], Ian D. Sharp [2✉] & Yanbo Li[1✉]

While self-healing is considered a promising strategy to achieve long-term stability for oxygen evolution reaction (OER) catalysts, this strategy remains a challenge for OER catalysts working in highly alkaline conditions. The self-healing of the OER-active nickel iron layered double hydroxides (NiFe-LDH) has not been successful due to irreversible leaching of Fe catalytic centers. Here, we investigate the introduction of cobalt (Co) into the NiFe-LDH as a promoter for in situ Fe redeposition. An active borate-intercalated NiCoFe-LDH catalyst is synthesized using electrodeposition and shows no degradation after OER tests at 10 mA cm$^{-2}$ at pH 14 for 1000 h, demonstrating its self-healing ability under harsh OER conditions. Importantly, the presence of both ferrous ions and borate ions in the electrolyte is found to be crucial to the catalyst's self-healing. Furthermore, the implementation of this catalyst in photoelectrochemical devices is demonstrated with an integrated silicon photoanode. The self-healing mechanism leads to a self-limiting catalyst thickness, which is ideal for integration with photoelectrodes since redeposition is not accompanied by increased parasitic light absorption.

[1] Institute of Fundamental and Frontier Sciences, University of Electronic Science and Technology of China, 610054 Chengdu, China. [2] Walter Schottky Institut and Physik Department, Technische Universität München, Am Coulombwall 4, 85748 Garching, Germany. [3] Institute of Engineering Innovation, The University of Tokyo, Tokyo 113-8656, Japan. [4] Engineering Research Center of Advanced Functional Material Manufacturing of Ministry of Education, Zhengzhou University, 450001 Zhengzhou, China. [5] Office of University Professors, The University of Tokyo, Tokyo 113-8656, Japan. [6] Research Initiative for Supra-Materials (RISM), Shinshu University, Nagano 380-8553, Japan. ✉email: sharp@wsi.tum.de; yanboli@uestc.edu.cn

Converting intermittent renewable energy resources, such as solar and wind energy, into storable chemical fuels is a key pathway toward a sustainable energy future[1–5]. Two principal routes currently under intensive investigation are electrocatalytic water splitting and carbon dioxide ($CO_2$) reduction. Hydrogen and hydrocarbons generated through these processes could serve as energy carriers for long-term storage, provide renewable transportation fuels, and enable versatile generation and distribution. Although these value-added chemicals are produced from water or $CO_2$ reduction reactions taking place at the cathodes, anodic hydroxyl ion oxidation is required to provide an abundant source of electrons and protons and complete the overall reaction. Therefore, as researchers pursue high-performance catalysts for the hydrogen evolution reaction (HER) and $CO_2$ reduction reaction ($CO_2$RR), it is equally important to develop efficient and durable catalysts for the kinetically challenging oxygen evolution reaction (OER)[6]. In the past decade, tremendous efforts have been devoted to this field and the activity of OER catalysts has been significantly improved[7–10]. Among these active OER catalysts, nickel–iron layered double hydroxides (NiFe-LDH) have stood out as one of the most promising candidates owing to their earth-abundant compositions and high OER activity in alkaline conditions[11,12]. Even though there has been debate over the role of Fe as either an active center or as a Lewis acid that enhances Ni activity[13], the ability of Fe to promote the OER activity of Ni oxyhydroxides is generally acknowledged. Recent mechanistic studies strongly suggest that iron comprises the active centers for OER in NiFe-LDH, while the nickel hydroxide lattice provides a thermodynamically stable host to accommodate the catalytic Fe centers[14–16]. The Pourbaix diagram of the Ni-water system[17] shows that the Ni-based compounds, including $Ni(OH)_2$ and $NiOOH$, are indeed thermodynamically stable under the OER conditions. However, the high valence state active Fe intermediates, such as cis-dioxo-Fe(VI) ($FeO_4^{2-}$)[14], have been found to be thermodynamically unstable under the OER conditions, consistent with the reported Pourbaix diagram of the Fe-water system[18]. These catalytic Fe centers may leach out of the catalyst during reaction[19,20], which leads to the degradation of the OER activity over time. To achieve ultimate stability of the catalyst, it has been suggested that self-healing of the leached catalytic centers may be the only realistic strategy[21,22]. Although promising short-term (1 h) dynamic stability of Fe active centers in NiFe oxyhydroxides has been recently demonstrated via energetically tuned surface adsorption of Fe from electrolyte[23], an effective self-healing strategy has not been realized for NiFe-LDH catalyst for long-term OER operation under harsh alkaline conditions. Thus, long-term stability of these catalysts represents a key property gap that must be overcome for the realization of practical systems incorporating this best-in-class OER catalyst.

The requirement for self-healing is that the leaching and redeposition of catalytic centers must reach dynamic equilibrium at OER operational potentials[24–27]. However, we found that for NiFe-LDH catalysts, the redeposition of the leached Fe catalytic centers at OER operational potentials was not effective during long-term operation. There exists a gap between the potentials for effective Fe redeposition and OER operational potentials, which is the major obstacle for realizing self-healing in this catalyst. Here, we propose to improve the efficiency of Fe redeposition at the OER operational potentials by employing Co as a catalyst for the oxidative redeposition of Fe hydroxide. Using this concept, a Co-catalyzed self-healing mechanism is proposed and a borate-intercalated NiCoFe-LDH (denoted as NiCoFe-$B_i$) catalyst is designed and experimentally determined to show excellent self-healing ability under alkaline conditions while retaining high activity. The benign synthesis condition, the self-healing ability at low operational potentials, and the self-limiting thickness of the NiCoFe-$B_i$ catalyst also make it ideally suited for integration into photoelectrochemical (PEC) systems for solar fuels generation. As a proof of concept, a self-healing water splitting photoanode is demonstrated with an integrated NiCoFe-$B_i$/NiO/CuO$_x$/n-Si photoanode, confirming that simultaneously stable and efficient performance can be obtained under alkaline conditions.

## Results and discussion

**Why self-healing is not achieved with NiFe-LDH catalysts.** Recently, it has been reported that short-term dynamic stability of Fe active centers in Fe-containing oxyhydroxides for OER can be achieved by adding Fe(III) ions into the KOH electrolyte[23]. In this approach, it is thought that the leaching of Fe active centers is balanced by the readsorption of Fe, which is governed by the adsorption energy between Fe(III) and other metal species in the catalysts. However, we find that catalyst stabilization based on this adsorption mechanism alone is not sufficient to maintain the stability of NiFe-LDH catalysts during long-term operation. To investigate this, borate-intercalated NiFe-LDH (denoted as NiFe-$B_i$) catalysts were electrodeposited on fluorine-doped tin oxide (FTO) glass substrates in borate buffer containing Ni(II) and Fe(II) ions. The OER performance of the catalysts was tested using a three-electrode electrochemical cell (Supplementary Fig. S1). The long-term stability of the NiFe-$B_i$ catalysts was assessed at a constant current density of 10 mA cm$^{-2}$ in different electrolytes (Supplementary Fig. S2). In particular, catalyst stabilization strategies were explored to compensate the losses of Fe catalytic centers and the intercalated borate ions ($B_4O_5(OH)_4^{2-}$) by adding Fe(II) ions and potassium borate ($K_2B_4O_7\cdot 4H_2O$, K$B_i$) into the KOH electrolyte. Consistent with the findings of Chung et al.[23], the stability of the NiFe-LDH catalyst was indeed improved in electrolyte containing both Fe(II) ions and borate ions (Supplementary Fig. S2). However, while dynamic stability seemed to be achieved for the first 12 h of testing (Fig. 1a), a gradual increase in the overpotential was observed after sustained long-term operation (Fig. 1a, b). Although the increase in overpotential was small, it indicated the catalyst was not ultimately stable and the degradation would eventually lead to the deterioration of catalytic activity over an extended period of time. The results indicate that self-healing of the NiFe-LDH catalyst cannot be achieved by simply adding Fe ions into the electrolyte. This could be ascribed to a reduced efficiency for Fe redeposition with increasing time due to the decreasing concentration of Fe ions in the electrolyte associated with gradual precipitation of Fe ions to Fe oxyhydroxides with low solubility[28–30]. We found by inductively coupled plasma mass spectrometry (ICP-MS) that although 50 μM Fe ions were added to the 1 M KOH electrolyte, the concentration of soluble Fe ions in the freshly prepared electrolyte was only 16.7 μM, which drastically decreased to 2.6 μM after aging in air for 100 h (Supplementary Table S1). Understanding this temporal change of Fe ion concentration is essential to the design of an effective self-healing strategy for long-term operation. While most efficient NiFe-LDH catalysts show OER operational potentials below ~1.6 V vs. reversible hydrogen electrode (RHE) at a current density of 10 mA cm$^{-2}$ (ref. [11]), the redeposition of leached Fe catalytic centers was found to be ineffective at these potentials, especially when the concentration of Fe ions in the electrolyte becomes extremely low during long-term operation.

To determine the potential range over which efficient Fe redeposition can be promoted, in situ measurement of Fe deposition rate was carried out by electrochemical quartz crystal microgravimetry (EQCM) (Supplementary Method 1). The electrolyte was aged in air for ~100 h after adding the Fe(II)

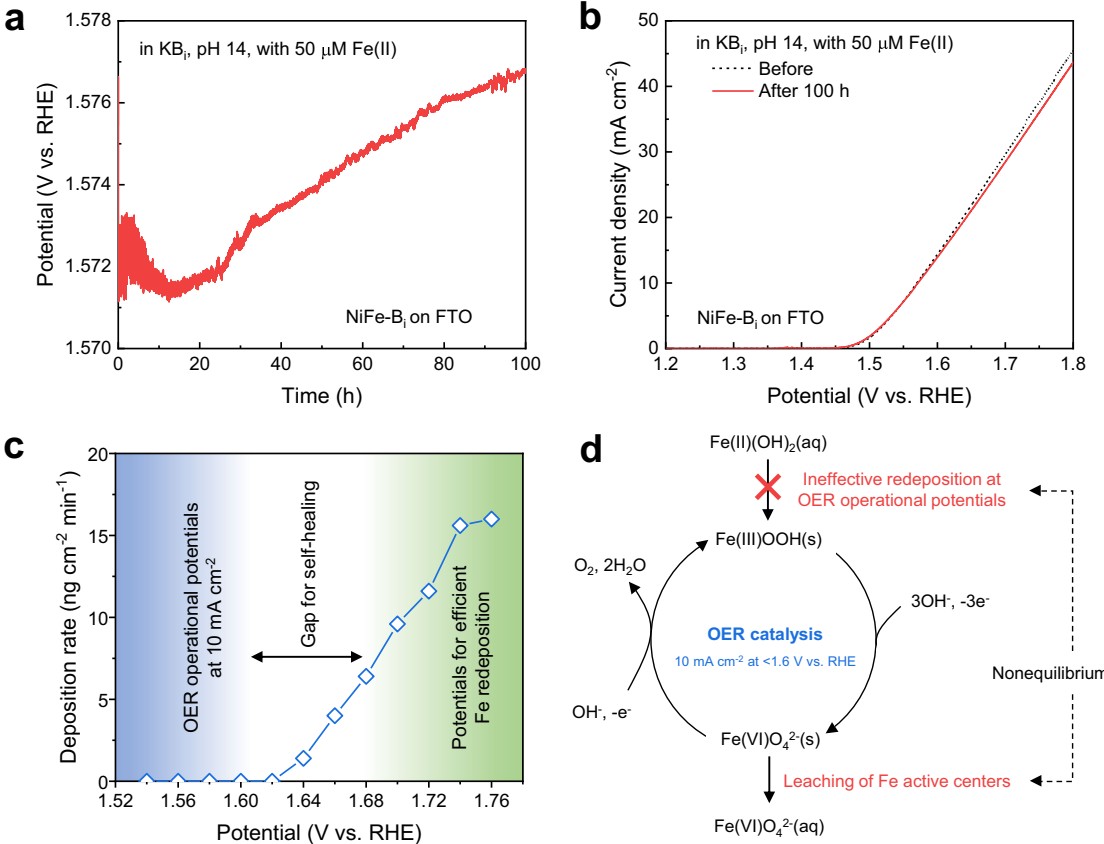

**Fig. 1 Possible reason for the failure of self-healing in NiFe-LDH catalyst. a** Stability test of the NiFe-B$_i$ catalyst measured at constant current density of 10 mA cm$^{-2}$ for 100 h in KB$_i$ electrolyte at pH 14 with 50 μM Fe(II) ions. **b** The OER polarization curve of the NiFe-B$_i$ catalyst before and after the stability test. The curves were measured at 1 mV s$^{-1}$ scan rate, without iR correction. **c** Deposition rates of Fe hydroxides on EQCM sensors at different potentials in Fe-containing KB$_i$ electrolyte. The gap between the blue and green regions illustrates the mismatch between the OER operational potentials and the potentials required for efficient Fe redeposition. **d** Proposed mechanism for the failure of self-healing in Fe-based catalysts. The OER catalysis is simplified by considering FeO$_4^{2-}$ as a representative active intermediate that is not thermodynamically stable and could leach out of the catalyst.

ions to best mimic the conditions of the electrolyte during long-term OER tests. The concentration of soluble Fe ions in the KB$_i$ electrolyte decreased from 22.7 μM in freshly prepared electrolyte to 7.5 μM after aging when 50 μM Fe(II) ions were initially added (Supplementary Table S1). The deposition rates of Fe hydroxides revealed by in situ EQCM measurement showed that there was almost no deposition below 1.62 V vs. RHE. Higher potentials of ~1.7 V vs. RHE were required for effective Fe deposition. Clearly, there exists a gap between the potentials for effective Fe redeposition and OER operational potentials, as illustrated in Fig. 1c. This potential gap prevented the redeposition of Fe at OER operational potentials and, hence, the dynamic equilibrium required for self-healing was not achieved. Figure 1d illustrates the proposed mechanism for the observed failure of NiFe-LDH OER catalysts to exhibit self-healing characteristics. The thermodynamic instability of the Fe(VI)O$_4^{2-}$ intermediate leads to the leaching of Fe active sites from the catalyst. Self-healing was not achieved due to the higher rate of leaching compared to Fe redeposition at the low OER operational potentials.

**The design and demonstration of self-healing NiCoFe-B$_i$ catalyst.** The above analysis shows that the key to realizing self-healing in the NiFe-LDHs catalyst is to improve the Fe redeposition efficiency at OER operational potentials. The redeposition of Fe occurs through the oxidation of Fe(II) ions dissolved in the electrolyte into Fe(III)OOH. Even though the Fe(II)/Fe(III) redox

potential is relatively low (0.771 V vs. RHE), a previous study on the kinetics of electrochemical Fe(II)/Fe(III) oxidation has revealed that no oxidation of Fe(II) occurs at oxygen evolution potentials on an anode material (lead) without the introduction of a catalytic effect[31]. Rather, such a catalytic effect was required for effective Fe(II)/Fe(III) oxidation at lower potentials[32]. It has been discovered that some ions, such as Cu, Mn and Co, have a catalytic effect on the Fe(II)/Fe(III) oxidation[33]. Here, we hypothesize that the catalytic effect of Co(II) ions could improve the Fe redeposition efficiency at the OER operational potential, so that the dynamic equilibrium required for self-healing can be reached. Based on this idea, we propose to introduce Co into the NiFe-LDH lattice as a promoter for the self-healing of Fe catalytic centers. Therefore, a NiCoFe-B$_i$ catalyst was designed, and its schematic structure is shown in Fig. 2a. The basal layers of the LDH structure consist of mixed [FeO$_6$], [CoO$_6$], and [NiO$_6$] octahedral sites. It is expected that the [NiO$_6$] octahedra provide a thermodynamically stable host to tightly bind the [FeO$_6$] and [CoO$_6$] octahedra[14]. Likewise, it is reported that bulky borate ions between the basal layers increase the interlayer distance, which is expected to improve the catalytic activity by enhancing the mass transport and enabling sites within the volume of the catalyst, rather than strictly at its surface, to be accessible for catalysis[34].

The NiCoFe-B$_i$ catalysts were deposited on FTO or gold (Au) substrates in borate buffer containing Ni(II), Co(II), and Fe(II) ions by oxidative electrodeposition[35]. Flat substrates were used for better understanding of the intrinsic catalytic

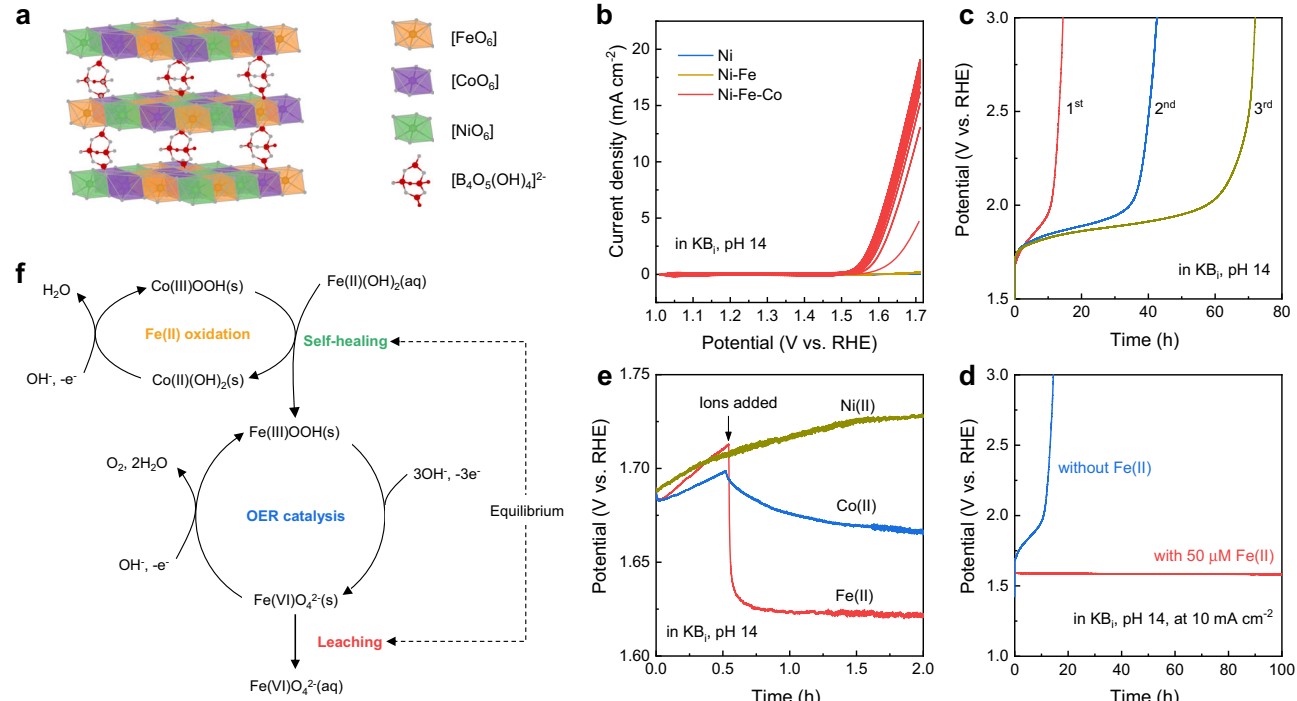

**Fig. 2 The demonstration of a NiCoFe-B$_i$ catalyst with self-healing ability. a** Schematic structure of the NiCoFe-B$_i$ catalyst. The basal layers are composed of mixed [FeO$_6$], [CoO$_6$], and [NiO$_6$] octahedral sites and are intercalated with [B$_4$O$_5$(OH)$_4$]$^{2-}$ ions. **b** In situ sequential deposition of metal oxyhydroxides on FTO substrate. 50 μM of Ni(II), Fe(II), and Co(II) ions were sequentially added to the KB$_i$ electrolyte at pH 14. After the addition of each ion, 20 cyclic voltammetry (CV) scans were performed between 1.01 and 1.71 V vs. RHE with a scan rate of 50 mV s$^{-1}$. **c** Chronopotentiometric curves of three NiCoFe-B$_i$ catalyst films measured sequentially in the same KB$_i$ electrolyte at pH 14 under constant current density of 10 mA cm$^{-2}$. **d** Comparison of the stability of NiCoFe-B$_i$ catalyst in KB$_i$ electrolyte with and without adding Fe(II) ions. **e** Chronopotentiometric curves of NiCoFe-B$_i$ catalyst films measured in KB$_i$ electrolyte at pH 14 with 30 μM Ni(II), Co(II), or Fe(II) ions added after 0.5 h, respectively. **f** The proposed Co-catalyzed self-healing mechanism of the NiCoFe-B$_i$ catalyst.

properties of the deposited catalyst layer, which has a thickness of approximately 35 nm (Supplementary Fig. S3). During electrodeposition of the catalyst layer, the catalytic effect of Co on the deposition of Fe (and Ni) was directly observed (Supplementary Fig. S4). Compared with NiFe-B$_i$ catalyst deposited at the same current density or at the same potential, the molar surface density of Fe (and Ni) in the NiCoFe-B$_i$ catalyst was significantly increased in the presence of Co(II) in the electrolyte. This finding lends strong support to the hypothesis that Co incorporation in the catalyst layer itself may promote Fe redeposition and thus allow self-healing for highly durable catalysts, as described below.

The catalytic effect of Co on the deposition of Fe was further supported by a series of in situ sequential electrodeposition experiments. As shown in Fig. 2b and Supplementary Fig. S5a–c, the deposition of Fe active sites was not successful in the potential range of 1.01–1.71 V vs. RHE in the absence of Co(II) ions. The drastically enhanced OER activity after adding Co(II) ions suggests Fe was successfully incorporated, which is ascribed to the catalytic effect of Co on the oxidation of Fe(II)(OH)$_2$ to Fe(III)OOH. The catalytic effect of Co, which is consistent with previous observations of the oxygenation of ferrous iron[33], bridges the gap between the deposition potential of Fe oxyhydroxides and the operational potential for OER, thus making self-healing possible. Our in situ sequential electrodeposition results are also consistent with previous findings which reveal that Fe, rather than Ni, is the active site for OER[14–16], since the incorporation of Ni after the Co-catalyzed Fe deposition has no effect on the OER activity (Supplementary Fig. S5b, e). More detailed discussion on the roles of Ni, Co and Fe in the NiCoFe-B$_i$

catalyst revealed by the in situ sequential electrodeposition experiments are giving in Supplementary Note to Fig. S5.

The self-healing effect was clearly observed when comparing the stability of the NiCoFe-B$_i$ catalyst in KB$_i$-containing electrolyte (at pH 14) with and without Fe(II) ions. In KB$_i$ electrolyte (0.25 M KB$_i$, pH 14) without the addition of Fe(II) ions, the OER activity of the NiCoFe-B$_i$ catalyst quickly deteriorated (Fig. 2c), suggesting that the borate buffer significantly increased the leaching rate of the Fe catalytic centers. After chronopotentiometric testing of the NiCoFe-B$_i$ catalyst layer at 10 mA cm$^{-2}$ for 10 h in KB$_i$ electrolyte, the molar surface density of Fe decreased from 10.4 to 5.4 nmol cm$^{-2}$ according to ICP-MS analysis. The higher leaching rate is also consistent with the higher solubility of Fe ions in KB$_i$ electrolyte as compared to KOH electrolyte (Supplementary Table S1). However, the self-healing effect of Fe ions was pronounced in this experiment. As shown in Fig. 2c, for the first sample tested in freshly prepared KB$_i$ electrolyte, the NiCoFe-B$_i$ catalyst degraded catastrophically after about 10 h. However, when the second and third samples were tested in the same electrolyte, the time for the catastrophic degradation was delayed to 35 and 60 h, respectively. This suggested that even the extremely small amount of leached Fe ions from the electrode into the electrolyte could slow down the degradation process of the NiCoFe-B$_i$ catalyst. By intentionally adding 50 μM Fe(II) ions into the electrolyte, the stability of the NiCoFe-B$_i$ catalyst was strikingly increased relative to that in KB$_i$ electrolyte without adding Fe(II) ions (Fig. 2d), suggesting the effectiveness of the self-healing even under conditions that drive a high leaching rate. As Fe impurities are already present in the electrolyte of commercial alkaline water electrolyzers due to the

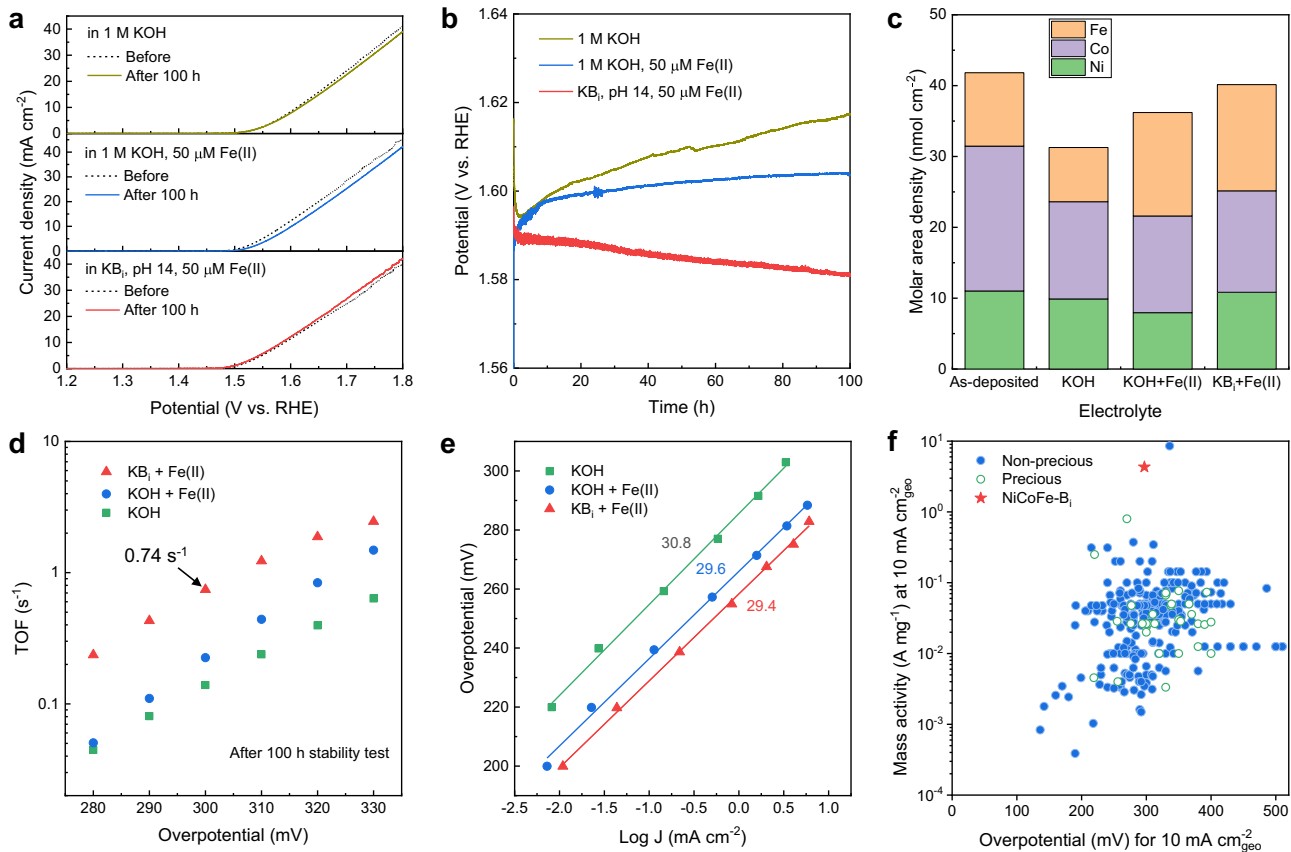

**Fig. 3 Self-healing and intrinsic catalytic properties of the NiCoFe-B$_i$ catalyst deposited on FTO substrate. a** The OER polarization curves of NiCoFe-B$_i$ catalysts before and after stability testing for 100 h in different electrolytes. The curves were measured at 1 mV s$^{-1}$ scan rate, without iR correction. **b** Chronopotentiometric curves of NiCoFe-B$_i$ catalysts measured at constant current density of 10 mA cm$^{-2}$ for 100 h in different electrolytes. **c** Quantitative ICP-MS analysis of the NiCoFe-B$_i$ catalysts before and after the 100 h stability tests in different electrolytes. **d** The oxygen evolution TOF of the NiCoFe-B$_i$ catalyst in different electrolytes. The values were calculated using polarization curves and ICP-MS data measured after the 100 h stability test. **e** Tafel plots of the NiCoFe-B$_i$ catalyst in different electrolytes. **f** Comparison of the mass activity of our NiCoFe-B$_i$ catalyst with literature-reported values[40].

adventitious Fe in KOH and the corrosion of stainless steel components[36,37], commercial alkaline water electrolyzers are expected to be tolerant to the addition of small concentration of Fe(II) ions into the electrolyte.

The influence of Ni(II) and Co(II) ions added into the electrolyte were also investigated. After testing the NiCoFe-B$_i$ catalyst films in freshly prepared KB$_i$ electrolytes for half an hour, 30 μM of Ni(II), Co(II), or Fe(II) ions were added to the electrolyte, respectively (Fig. 2e). The addition of Ni(II) ions did not prevent the degradation of the catalytic activity, from which it is concluded that they do not participate in the self-healing mechanism. While both Co(II) and Fe(II) ions were found to improve the catalytic activity of the samples, the causes for the improvement were found to be different. In particular, it was observed that the catalyst film thickness increased after the chronopotentiometry test in Co(II)-containing electrolyte, as confirmed by the significant optical darkening of the sample and scanning electron microscopy (SEM) observation (Supplementary Fig. S6). As such, the improved catalytic activity is assigned to the increased loading of Co which also possesses moderate OER activity. In contrast, the morphology of the catalyst was unchanged after the chronopotentiometry test in Fe(II)-containing electrolyte, indicating that the improved catalytic activity was not due to the thickening of catalyst film, but truly due to self-healing.

The stability of the NiCoFe-B$_i$ catalysts was tested in different electrolytes and the molar surface density within the films before

and after testing was analyzed by ICP-MS of subsequently digested layers (Fig. 3a–c). In pure 1 M KOH electrolyte, the potential increased by ~23 mV after 100 h at a constant current density of 10 mA cm$^{-2}$ (Fig. 3b). The degradation of OER activity was mainly due to the loss of Fe active centers during the OER process, as evidenced by the reduced molar surface density of Fe after the stability test (Fig. 3c). When Fe(II) ions were added to the KOH electrolyte, the molar surface density of Fe increased after the stability test, suggesting that the self-healing mechanism of Fe redeposition was activated. However, the galvanostatic operating potential still increased by ~14 mV after 100 h (Fig. 3b), which was most likely due to the loss of intercalated borate ions, as evidenced by the reduced intensity of the B 1 s core-level X-ray photoelectron spectroscopy (XPS) peak after the stability test (Supplementary Fig. S7). Furthermore, the ICP-MS data showed that the molar surface density of Ni was reduced after the stability test in Fe(II)-containing KOH electrolyte. Although Ni hydroxide is thermodynamically stable under the OER condition, the loss of intercalated borate ions could affect the structural integrity of the catalyst. The drastic structural rearrangement of α-Ni(OH)$_2$ to γ-NiOOH during OER could cause the destabilization and leaching of otherwise thermodynamically stable Ni species[15]. By further adding borate ions into the electrolyte to compensate the loss of intercalated borate ions (Supplementary Fig. S8), the degradation of the catalytic activity was eliminated, as shown in Fig. 3a,b. Indeed, the catalytic activity was even observed to gradually improve during the OER process, which is ascribed to the

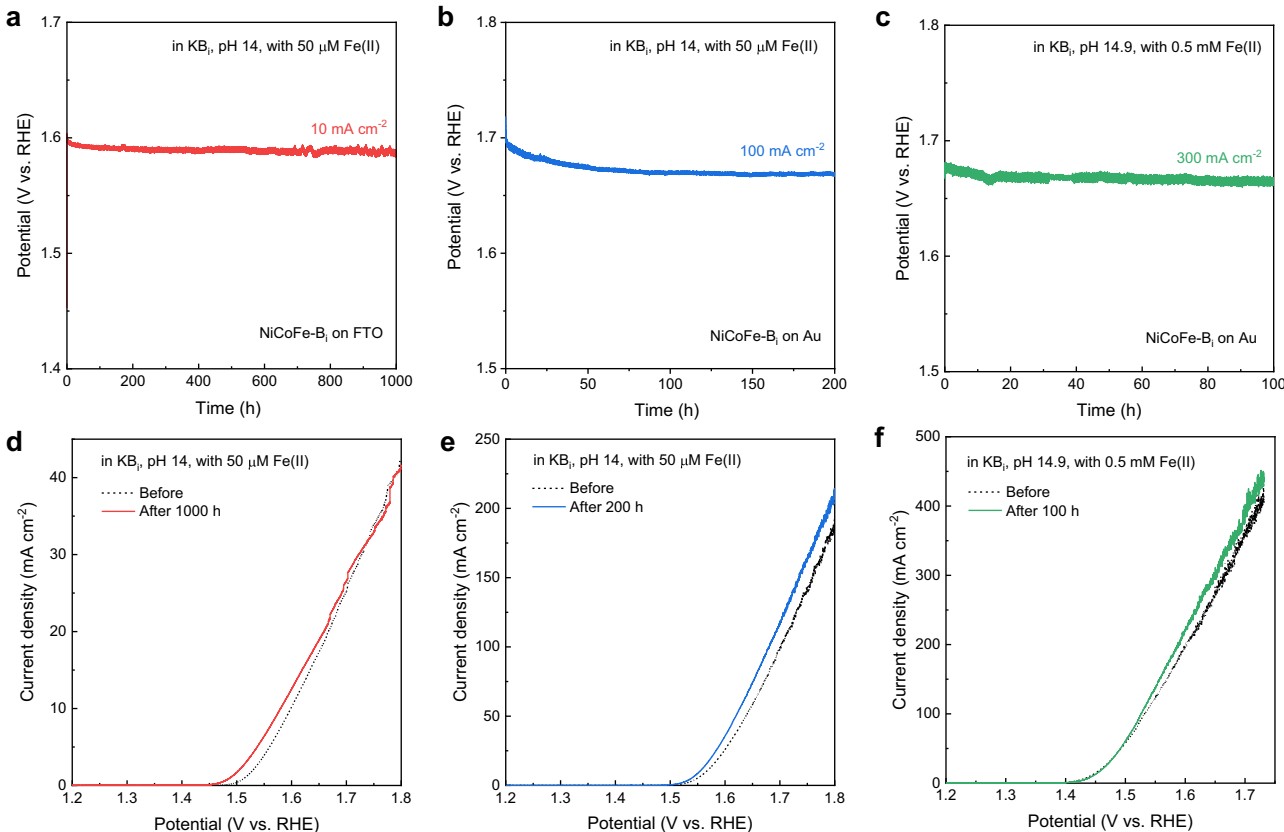

**Fig. 4 Self-healing ability of the NiCoFe-B$_i$ catalyst under various harsh conditions.** Chronopotentiometry tests of the NiCoFe-B$_i$ catalysts (**a**) on FTO substrate at 10 mA cm$^{-2}$ for 1000 h in KB$_i$ electrolyte at pH 14, (**b**) on Au substrate at 100 mA cm$^{-2}$ for 200 h in KB$_i$ electrolyte at pH 14, and (**c**) on Au substrate at 300 mA cm$^{-2}$ for 100 h in KB$_i$ electrolyte at pH 14.9. The corresponding polarization curves before and after the stability tests are plotted in (**d–f**). All the polarization curves were measured with 1 mV s$^{-1}$ scan rate and all potentials were not corrected for iR losses.

increased Fe catalytic centers incorporated during the self-healing process, as indicated by the ICP-MS results (Fig. 3c). The morphological change of the NiCoFe-B$_i$ catalyst during the stability test in Fe(II)-containing KB$_i$ electrolyte is presented in Supplementary Fig. S9, which indicates the activation of the self-healing mechanism during the OER process.

The stability of the NiCoFe-B$_i$ catalyst was also tested in KB$_i$ electrolyte (pH 14) containing 50 μM Fe(III) ions instead of the Fe(II) ions (Supplementary Fig. S10). The long-term stability of the catalyst was not achieved in Fe(III)-containing electrolyte. The redeposition of Fe in this case is mainly governed by the adsorption of Fe(III) on the catalyst[23], which is not sufficient to maintain the stability of the catalyst during long-term OER operation. These results suggest that the presence of Fe(II) ions in the electrolyte is essential for achieving self-healing of the NiCoFe-B$_i$ catalyst. To track the change of concentrations of Fe(II) ions during the course of the long-term OER test, a colorimetric method for the determination of Fe(II) in the presence of Fe(III) was used[38,39] (Supplementary Method 2). It was found that although the majority of the Fe(II) ions added to the KB$_i$ electrolyte were oxidized into Fe(III), a small fraction (~1.3 μM) of Fe still existed in the form of Fe(II) in the electrolyte after the 100 h OER test (Supplementary Figs. S11, S12).

Additional experiments were carried out to clarify the roles of Ni (Supplementary Fig. S13) and borate ions (Supplementary Figs. S14, S15) in the catalyst and electrolyte. From these combined results of systematic electrochemical measurements, we can construct a portrait of the roles of each element in the NiCoFe-B$_i$ catalyst, as summarized in Supplementary Table S2. The proposed mechanism for the self-healing of the NiCoFe-B$_i$

catalyst during the OER process, derived from these observations and the known activity of Co for catalyzing Fe(II) to Fe(III) oxidation, is shown in Fig. 2f. The leaching of Fe active centers is compensated by the Co-catalyzed redeposition of Fe oxyhydroxide. As indicated by the robust operational characteristics, as well as the compositional stability assessed by ICP-MS, the leaching and redeposition reach dynamic equilibrium, and therefore, the requirement for self-healing is satisfied. Furthermore, self-healing property was also achieved with a NiMnFe-B$_i$ catalyst by replacing the Co with Mn that also exhibits a catalytic effect for Fe(II)/Fe(III) oxidation[33], demonstrating the universality of the catalyzed self-healing mechanism (Supplementary Fig. S16).

**The intrinsic catalytic properties of NiCoFe-B$_i$ catalyst**. To assess the catalytic activity of the NiCoFe-B$_i$ catalyst and compare it to state-of-the-art alkaline OER catalysts, turnover frequencies (TOFs), Tafel slopes, and mass activities were quantified. A lower-limit value (by considering all the NiCoFe atoms as the active sites) for the oxygen evolution TOF was estimated to be 0.74 s$^{-1}$ at an overpotential of 300 mV (with iR correction, see Supplementary Fig. S17) for the stabilized NiCoFe-B$_i$ catalyst (Fig. 3d), which is comparable to that of the highest performing NiFe-LDH catalysts[40]. The TOF values were lowered to 0.14 and 0.22 s$^{-1}$ for samples tested in pure KOH and KOH with Fe(II) ions due to the loss of Fe catalytic centers and intercalated borate ions, respectively. Despite the difference in TOF values in different electrolytes, the Tafel slopes were all found to be ~30 mV/decade for the NiCoFe-B$_i$ catalyst (Fig. 3e). A Tafel slope of ~30 mV/decade is indicative of an equilibrated two-electron

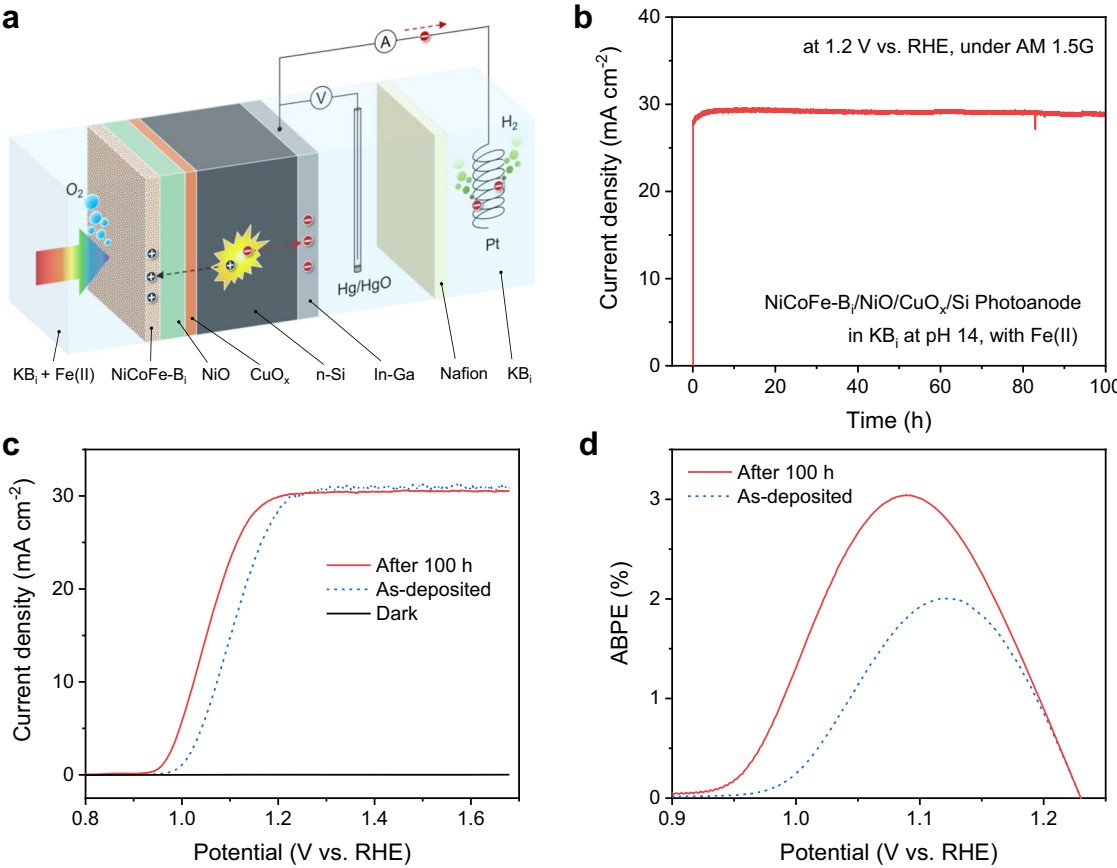

**Fig. 5 PEC performance of an integrated NiCoFe-B$_i$/NiO/CuO$_x$/n-Si photoanode. a** Schematic illustration of the PEC device. **b** Chronoamperometric curve of the photoanode measured at 1.2 V vs. RHE under AM 1.5 G for 100 h. **c** *J–V* curves of the photoanode before and after the stability test under AM 1.5G and in the dark. **d** ABPE curves of the photoanode before and after the stability test.

process that precedes the rate determining step[41], which was not affected by adding Fe(II) and borate ions to the electrolyte. The mass activity of the NiCoFe-B$_i$ catalyst was measured to be 4.32 A mg$^{-1}$ at a geometric current density of 10 mA cm$^{-2}$, which is the highest value for OER catalysts with overpotentials below 300 mV (Fig. 3f). The high mass activity makes the NiCoFe-B$_i$ catalyst ideal for integrated PEC water splitting devices[42], as an ultrathin catalyst layer is sufficient to achieve high OER activity, thus minimizing its light-shading effect (Supplementary Fig. S18). The amount of O$_2$ evolved during the self-healing OER process was also quantified by gas chromatography (GC), obtaining nearly unity Faradaic efficiency for the O$_2$ evolution (Supplementary Fig. S19).

**Self-healing ability under harsh OER conditions**. The self-healing ability of the NiCoFe-B$_i$ catalyst under operation for long times, at high current densities, and under extreme environments were tested, as shown in Fig. 4. Under extended stability testing at a current density of 10 mA cm$^{-2}$ at pH 14, the catalyst showed no sign of degradation after 1000 h (Fig. 4a). When the current density was increased to 100 mA cm$^{-2}$ using a more conductive Au substrate, the catalyst still showed no sign of degradation after 200 h (Fig. 4b). We further tested the stability of the NiCoFe-B$_i$ catalyst in highly concentrated alkaline electrolyte (0.25 M KB$_i$ + 9.5 M KOH, pH ~14.9) at a current density of 300 mA cm$^{-2}$ for 100 h (Fig. 4c). The catalyst was found to be stable against this extremely harsh OER environment. In addition, in all cases an initial improvement in catalytic activity was observed before eventual stabilization. The maximum concentration of KOH tested here

(9.5 M, ~39 wt.%) is higher than those used in commercial alkaline water electrolyzers (5.4–8.2 M, 25–35 wt.%)[43]. In state-of-the-art CO$_2$RR catalysis systems, highly concentrated KOH electrolyte (1–10 M, 5.3–40 wt.%) is also commonly used to fine-tune the local reaction environment for an improved C–C coupling[44–46]. Our results demonstrate that the NiCoFe-B$_i$ catalyst is robust against similarly harsh environments due to its extraordinary self-healing ability and may find practical applications in alkaline water electrolysis and CO$_2$RR electrolysis.

**PEC application of the NiCoFe-B$_i$ catalyst**. The Co-catalyzed self-healing mechanism has led to a unique property of the NiCoFe-B$_i$ catalyst—the thickness of the catalyst film does not increase even with excess Fe(II) ions in the electrolyte (Fig. 2e and Supplementary Fig. S6e). Because the redeposition of Fe is catalyzed by Co, the Fe atoms can only be deposited on sites adjacent to the Co sites. Thus, the deposition of pure Fe hydroxide overlayers on top of the catalyst film is prohibited. In contrast, for Co-based self-healing catalysts (e.g., Co-Pi, Co-B$_i$) the thickness of the catalysts increases continuously during the self-healing process with excess Co(II) ions in the electrolyte[24,26]. The unique self-limiting thickness of the NiCoFe-B$_i$ catalyst discovered here makes such catalysts particularly suitable for PEC applications, as a high light transmittance through the catalyst layer can be maintained during the self-healing OER process. To demonstrate the applicability of the self-healing catalyst in PEC applications, an integrated NiCoFe-B$_i$/NiO/CuO$_x$/n-Si photoanode was fabricated (Fig. 5a), in which a CuO$_x$ interlayer was employed to improve the interface energetics of the p-NiO/n-Si heterojunction. At a bias of

1.2 V vs. RHE, the photocurrent increased steadily for the first few hours and remained constant at approximately 29 mA cm$^{-2}$ for the remainder of the 100 h test (Fig. 5b). The photocurrent density-potential (J-V) curves in Fig. 5c reveal a cathodic shift of the onset potential by ~50 mV after the stability test, which is attributed to the improved OER activity of the NiCoFe-B$_i$ catalyst due to self-healing. As a result, the applied bias photon-to-current efficiency (ABPE) of the photoanode was improved from 2.0% to 3.0% (Fig. 5d). The efficiency and stability of the photoanode compare favorably with previously reported n-Si photoanodes (Supplementary Fig. S20). The self-improved PEC performance of the photoanode demonstrates that the NiCoFe-B$_i$ catalyst is ideally suited for PEC applications.

This work presents a self-healing strategy to address the long-term stability issue of NiFe-LDH catalyst due to the leaching of Fe catalytic centers. We found that in its usual configuration, self-healing of NiFe-LDH catalyst was not possible because catalytic Fe could not be redeposited at OER operational potentials. We proposed and investigated the introduction of Co as a promoter for in situ redeposition of Fe. A highly active borate-intercalated NiCoFe-LDH catalyst was synthesized using a simple electro-deposition method and showed no sign of degradation after OER tests at 10 mA cm$^{-2}$ at pH 14 for 1000 h (or at 300 mA cm$^{-2}$ at pH 14.9 for 100 h), demonstrating its extraordinary self-healing ability under harsh OER conditions. The self-healing NiCoFe-B$_i$ catalyst we have realized for OER under highly alkaline conditions could not only find applications in commercialized alkaline water electrolyzers, but also in state-of-the-art CO$_2$RR electrocatalysis systems. Furthermore, it is nearly ideally suited for integration into PEC systems, as demonstrated with a self-improving Si photoanode. Perhaps more importantly, the mechanism proposed herein provides a general guideline for the development of self-healing catalysts by employing additional constituents specifically designed to promote the self-healing of the catalytic centers.

## Methods

**Preparation of electrodes**. FTO glass substrates were sequentially cleaned with precision detergent (Alconox), deionized water, acetone, and isopropanol by ultrasonication for 15 min each. To lower the contact resistance, 100 nm thick silver (Ag) films were deposited on the edges of the substrates by electron-beam evaporation (Angstrom Engineering AMOD) through a shadow mask. Then a copper wire was soldered onto the Ag contact using indium. The exposed metallic parts were then encapsulated with epoxy (Araldite). The exposed area of the FTO electrodes was approximately 1.1 cm$^2$. For Au electrodes, a copper wire was soldered onto the backside of Au foils (0.1 mm in thickness) using indium and then encapsulated with epoxy (Araldite). The exposed areas of the Au electrodes were 0.23 cm$^2$ for the one tested in Fig. 4b and 0.116 cm$^2$ for the one tested in Fig. 4c. The Au electrodes were etched with a mixture of HNO$_3$ and H$_2$SO$_4$ (1:3 in v/v) for 10 s and rinsed thoroughly with deionized water before electrodeposition of the catalyst.

**Electrodeposition of NiCoFe-B$_i$ and NiFe-B$_i$ catalyst**. NiCoFe-B$_i$ catalyst was electrodeposited onto FTO or Au substrates in potassium borate buffer containing Ni(II), Co(II) and Fe(II) ions. Potassium borate (K$_2$B$_4$O$_5$(OH)$_4$, KB$_i$) buffer with a concentration of 0.25 M was prepared by mixing 1 M boric acid (H$_3$BO$_3$, ACS grade, 99.5%, Aladdin) with 0.5 M potassium hydroxide (KOH, Greagent, 95%, Aladdin). The pH of the KB$_i$ buffer was further adjusted to 10 by adding KOH. Before electrodeposition, the buffer solution was deoxygenated by purging with Argon (Ar, 99.999%) gas for at least 10 min. Then, 0.5 mM cobalt(II) nitrate (Co(NO$_3$)$_2$·6H$_2$O, 99.99% metals basis, Aladdin), 2 mM nickel(II) sulfate (NiSO$_4$·6H$_2$O, 99.99% metals basis, Aladdin), and 0.8 mM iron(II) sulfate (FeSO$_4$·7H$_2$O, 99.95% metals basis, Aladdin) were sequentially added to the KB$_i$ buffer solution under magnetic stirring. The solution was under continuous Ar bubbling and magnetic stirring during electrodeposition. Electrodeposition was conducted with a potentiostat (BioLogic SP-200) in a three-electrode configuration using Ag/AgCl as the reference electrode and a Pt wire as the counter electrode. The NiCoFe-B$_i$ catalyst was deposited on an FTO substrate under a constant current density of 20 μA cm$^{-2}$ for 8 min. After electrodeposition, the electrode was rinsed thoroughly with deionized water before the OER test. NiFe-B$_i$ catalyst was deposited onto FTO substrates in a similar way to that used for NiCoFe-B$_i$ deposition. The electrolyte was 0.25 M KB$_i$ solution buffered at pH 10 containing 2 mM NiSO$_4$ and 0.8 mM

FeSO$_4$. Due to the low deposition rate at low current density, the electrodeposition of the NiFe-B$_i$ catalyst characterized in Fig. 1a was carried out at a constant current density of 1 mA cm$^{-2}$ for 8 min.

**OER characterization**. The OER performance of the catalysts was characterized with a potentiostat (BioLogic SP-200) in a three-electrode electrochemical cell using Ag/AgCl as the reference electrode and a Pt wire as the counter electrode (Supplementary Fig. S1). The anode chamber and the cathode chamber were separated by a proton-exchange membrane (PEM, Nafion 117, Dupont). This prevented the deposition of Fe on the counter electrode as well as the deposition of leached Pt on the working electrode. The cell was integrated with a water jacket and the temperature of the electrolyte was maintained at 20 °C using a constant temperature water circulator. The catalysts were tested either in 1 M KOH electrolyte (pH 14) or in 0.25 M KB$_i$ electrolyte (pH 14), with and without adding 50 μM FeSO$_4$. The 0.25 M KB$_i$ electrolyte was prepared by mixing 1 M H$_3$BO$_3$ with 0.5 M KOH, and the pH was adjusted to 14 by further adding approximately 1.5 M KOH. Highly concentrated electrolyte containing 0.25 M KB$_i$ and 9.5 M KOH (pH ~14.9) with 0.5 mM Fe(II) ions was also used for the OER test. Polarization curves of the catalysts were measured under anodic scan at a rate of 1 mV s$^{-1}$ under magnetic stirring. The stability of the catalysts was tested under current densities of 10, 100, or 300 mA cm$^{-2}$, respectively. The potentials vs. Ag/AgCl reference electrodes were converted into RHE using the Nernst equation:

$$E_{RHE} = E_{Ag/AgCl} + 0.197 + 0.0582*pH \qquad (1)$$

The TOF values of the catalysts were calculated using the polarization curves and the molar mass of catalysts measured after the 100 h stability test:

$$TOF(s^{-1}) = J/(4enN) \qquad (2)$$

where J (in A cm$^{-2}$) is the current density obtained from the polarization curves after iR corrections, e is the elemental charge, n (in mol cm$^{-2}$) is the total molar surface density of FeCoNi, and N is the Avogadro constant. The ohmic resistance (R) for iR corrections were determined by electrochemical impedance spectroscopy (EIS), as shown in Supplementary Fig. S17.

To determine the Tafel slopes, the applied potentials were varied in 20 mV increments and maintained until the current reached a steady-state value. The potentials were also corrected for iR losses.

Gas chromatography (GC) measurement of the evolved oxygen was carried out in a flow cell configuration. An air-tight electrochemical cell with gas inlet and outlet was used for the GC measurement. The cathode and anode were also separated with a PEM to prevent the back reaction of H$_2$ with O$_2$. The electrolyte in the anode chamber was purged with Ar gas for 20 min to remove oxygen dissolved in the electrolyte and in the head space. The volume of the head space was measured to be ~30 mL. The flow rate of the Ar gas was maintained at 10 sccm during the GC test using a mass flow controller (MFC, KOFLOC 8500). The concentration of O$_2$ in the head space of the chamber was analyzed by GC (Shimadzu GC-2014). After OER at a constant current for any given time, the maximum concentration of O$_2$ inside the cell can be calculated assuming 100% Faradaic efficiency (FE). The actual FE was then obtained by dividing the measured O$_2$ concentration with the calculated O$_2$ concentration at any given time.

**Si photoanode fabrication and characterization**. A single-side polished n-type Si <100> wafer with a thickness of 500 μm and resistivity of 0.7–0.9 Ω cm was cut into 10×10 mm pieces. The Si substrate was then cleaned with precision detergent, deionized water, acetone, and isopropanol by ultrasonication for 10 min each. Thin layers of CuO$_x$ (0.5 nm) and NiO (20 nm) were then sequentially deposited on the Si substrate by electron-beam evaporation (Angstrom Engineering AMOD) using Cu$_2$O (99.9% purity) and NiO (99.9% purity) pellets as the source materials. To form an Ohmic contact, the backside of the Si substrate was etched with 20% HF aqueous solution and In-Ga eutectic was then applied to the freshly-etched surface. The backside of the Si substrate was connected to a copper wire using Ag paste and the metallic parts were then completely encapsulated with epoxy. The NiCoFe-B$_i$ catalyst was photo-electrodeposited at 30 μA cm$^{-2}$ for 10 min in 0.25 M KB$_i$ solution at pH 10 with 0.5 mM Co(NO$_3$)$_2$, 2 mM NiSO$_4$, and 0.8 mM FeSO$_4$ under simulated sunlight illumination. The active area of the photoanode was 0.91 cm$^2$. PEC properties of the Si photoanode were tested in a three-electrode configuration using Hg/HgO as the reference electrode and a Pt wire as the counter electrode (Fig. 5a). The anolyte and catholyte, separated with a Nafion membrane, were 0.25 M KB$_i$ solution (pH 14) with and without 50 μM FeSO$_4$, respectively. To reduce light-scattering effects, the anolyte was filtered with a 5-μm capsule filter to remove the precipitated Fe hydroxide particles. Chronoamperometry testing was carried out at 1.2 V vs. RHE for 100 h under AM 1.5 G simulated sunlight illumination (SAN-EI Electronic, XES-40S3-TT). The current–potential curves were recorded with an anodic scan (10 mV s$^{-1}$) before and after the chronoamperometry test.

**Structural and elemental analysis of the catalysts**. The morphology of the catalysts was characterized using SEM (Zeiss Crossbeam 340). Cross-sectional

sample was prepared by focused ion beam etching (JEOL JIB-4600F). The cross-sectional STEM, EDS mapping, and HRTEM were performed with a JEOL JEM-2800 equipped with X-MAX 100TLE SDD detector (Oxford Instruments). The molar mass of the catalysts loaded on the substrate was quantified by ICP-MS (PerkinElmer NexION 350). For ICP-MS measurements, the catalysts on the FTO substrates were dissolved into 10 mL of 2 wt.% $HNO_3$ aqueous solution. The background concentrations of Ni, Co, and Fe ions in the $HNO_3$ solution were also measured and subtracted from the measured concentrations of the samples. XPS spectra were taken using a Thermo Scientific ESCALAB 250 Xi with a monochromatic Al Kα X-ray source with a beam size of 400 μm. The core-level spectra were collected with a constant analyzer energy of 50 eV and a step size of 0.05 eV and the binding energy was calibrated by setting the binding energy of the hydrocarbon C 1 s feature to 284.8 eV. Spectrum analysis was performed with the Thermo Scientific Avantage software. The transmission spectrum of the catalyst film deposited on FTO substrate was measured with a SHIMADZU UV-2600 spectrophotometer.

## Data availability

Data reported in the main article are provided in the Source Data file. The remaining data that support the findings of this study are available from the corresponding author upon request. Source data are provided with this paper.

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

## Acknowledgements

This work was supported by the National Natural Science Foundation of China (No. 21872019). I.D.S. acknowledges funding by the Deutsche Forschungsgemeinschaft (DFG, German Research Foundation) under Germany´s Excellence Strategy—EXC 2089/1—390776260 and by TUM.Solar in the context of the Bavarian Collaborative Research Project Solar Technologies Go Hybrid (SolTech). M.N., N.S., and K.D. acknowledge financial support by the Artificial Photosynthesis Project of the New Energy and Industrial Technology Development Organization (NEDO) and by the University of Tokyo Advanced Characterization Nanotechnology Platform in the Nanotechnology

Platform Project sponsored by the Ministry of Education, Culture, Sports, Science and Technology (MEXT), Japan (JPMXP09-A-21-UT-0046).

## Author contributions

Y.L. and I.D.S. proposed the project; Y.L., I.D.S., N.S. and K.D. supervised the project; Y.L. designed the experiments and carried out the PEC tests; M.N carried out the STEM and HRTEM experiments; C.F. carried out all the other experiments with the assistance from F.W., Z. L., Y.X., J.F., Q.Z., and Q.W.; C.C. and Y.H. discussed the results; Y.L. and I.D.S. analyzed the results and wrote the manuscript. All authors commented on and revised the manuscript.

## Funding

## Competing interests

The authors declare no competing interests.
