## [Peer Review File · Nature Communications]

A self-healing catalyst for electrocatalytic and photoelectrochemical oxygen evolution in highly alkaline conditionsREVIEWER COMMENTS

Reviewer #1 (Remarks to the Author):

The self-healing concept of transition metal OER electrocatalysts is an important area of study, however the underlining premise of this manuscript has been introduced by Chung et al. (ref 23). In this past work, iron (II) ions are added into the OER electrolyte to replenish iron into iron-M hydr(oxy)oxide electrocatalysts, where M includes both Ni and Co. This manuscript does describes this past work with NiFe, but does not refer to the improvement found with FeCo, which is particularly relevant to the authors' ternary FeNiCo-hydr(oxy)oxide electrocatalyst self-healing with similar iron(II) in the OER electrolyte. While the ternary shows even better improvement, this reviewer found the work lacking sufficient novelty for Nature Communications.

Additionally, the study used two different substrates (FTO and gold) for different aging conditions which calls into question the role of the substrate. Also, while the role of boric acid is cited to facilitate the dissolution and re-deposition of the iron, the chemistry is not clear.

Reviewer #2 (Remarks to the Author):

Manuscript NCOMMS-21-21000

"A self-healing catalyst for electrocatalytic and photoelectrochemical water oxidation in highly alkaline conditions" by Sharp, Li et al.

Recommendation: Minor revisions

General comment: Very interesting work about self-healing OER catalyst that is relatively stable (1000h at 10mAcm⁻²) in alkaline media. Shown results, including approach to catalyst design is of relevance for the field. Work is filling existing gap in the literature, discussing in very relevant manner stability of self-healing catalysts. Results are supporting conclusions and claims. Paper is well written with adequate graphic material. I could recommend acceptance in current form, however it would be better if authors deal with several comments I have included in the report:

More specific comments:

1) On page 3 lines 34-36, authors stated : "Although these value-added chemicals are produced from water or CO₂ reduction reactions taking place at the cathodes, water oxidation at anodes is required to provide an abundant source of electrons and protons and complete the overall reaction"

This is incorrect sentence. If you address reactions alkaline media (like in the title is mentioned) then HER is based on cathodic water reduction, however, OER is based on anodic hydroxyl ion oxidation, not on anodic water oxidation....water oxidation proceeds in acidic media..... So, please correct this.

2) Again, page 3 lines 40-43, authors talk about water oxidation in alkaline media...

Please make this consistent, according to comment 1.

3) On page 4 authors assume that Co will alter redox potential of Fe redeposition. How exactly ?

4) Page 4, line 71 authors claim extremely harsh alkaline conditions, however, 10mAcm⁻² is not even harsh condition.....I would definitely not use formulation "extremely harsh"

5) Authors claim self-limiting thickness.... This should be at very limited potential range....As

overpotential increases the larger will be potential difference between OER potential and redeposition potential for Fe. Is this wrong assumption or ...?

6) Page 5, line 91, authors claim intercalation of borate ions ($B_4O_5(OH)_4^{2-}$)...is this really possible ?

7) On page 6 authors claim opposite to my assumption in comment 6. From EQCM data no redeposition before 1.62 V was detected....It seems that directly oxide is redeposited, definitely not metal that gets oxidized....Could authors comment such positive potential of Fe deposition.? How that reaction looks like ?

8) On page 7, line 136, authors accept idea from Ref.33 that some cations (Co, Mn, Cu) catalyze Fe-deposition. They prove it by enhanced deposition of Fe in presence of Co. However, they introduce boron based anion (borate) to increase distance between basal planes....Is it really fact that borate ions get intercalated as stated in Re.34. Is there any proof for that ?

9) Authors used Ag/AgCl reference...this reference electrode should be avoided in alkaline-media....specially if concentration of base is above 0,1 M. Check for example <https://doi.org/10.1002/cssc.201702287>

10) Authors did not mention effective surface area due to gas-bubble blockage and how this phenomena influences their results.

Reviewer #3 (Remarks to the Author):

This work by C. Feng et al. describes the method for depositing and characterization of a "self-healing" OER catalyst containing Co, Fe, and Ni in a LDH structure with borate ions intercalated in the interlayer spacing. The self-healing properties are imparted by dissolved Fe(II) in the electrolyte solution which undergo catalytic reaction with Co sites in the thin film to redeposit Fe into the catalyst layer. The presence of borate ions in the electrolyte is also important for the catalyst's self-healing ability.

Overall, I found the paper to be well written, with clear hypothesis-driven experiments leading to conclusions that are well-supported by the data. The methods are sound, and the interpretation of the results is appropriate and conclusions/implications are not overstated or sensationalized.

The authors have done a nice job of tying together some loose ends in the field to show conclusively that the presence of Co in the LDH catalyst film is necessary to stabilize the dynamic Fe processes and maintain high activity of the materials. However, each of these items individually, (1) that Fe in solution is necessary for sustained performance, and (2) that ternary NiCoFe LDH catalysts are more stable than NiFe or CoFe, are already well known in the field. So while I am struggling to find faults with anything in the paper, I think the authors need to better emphasize what they think are the *particularly noteworthy, new results* here so that it is clear why this work merits publication in Nature Communications.

For example, the distinction that soluble Fe(II) rather than Fe(III) is necessary seems to be a new finding, and that including borate ions in the electrolyte is integral for halting performance degradation is also new, yet neither of these items are mentioned in the abstract. I think the authors need to clearly highlight what sets this work apart so that it doesn't read like just another iteration of making a new catalyst formulation and testing it in a PEC device, of which there are so many papers coming out every week in this very crowded field. I think there is a lot of novelty and important findings in here already, it is just a bit buried.

Otherwise, I have a very few minor comments/suggestions which are outlined below.

-page 8, line 148 - What is the mechanism of electrodeposition of the NiCoFe-Bi LDHs under anodic conditions? Is it known? Can the authors provide a reference or reasonable hypothesis?

-page 8, line 150 - The films are described as being "a few nanometers" thick. Can you be more specific with the film thickness, and how was the thickness determined? Were any substrate-induced effects observed for thin films? (i.e. was OER activity influenced by substrate for films of a given thickness?)

-page 11, line 220-222 - "The drastic structural rearrangement...." Have you measured the basal spacing (i.e. interlayer distance) in the NiCoFe-Bi catalyst? Or could you estimate it based on the size of the borate anion as depicted in your figures? It looks quite bulky compared to e.g. chloride or nitrate or sulfate, which are often the interlayer anions present during synthesis of these types of LDHs. I am wondering if the borate itself is so large that it prevents collapse/expansion of the structure during cycling and this is its contribution to enhancing the catalyst stability. One could attempt to synthesize the LDH with some other large anion in the interlayer to see if a similar effect is observed. Or perhaps monitor changes in the structure during cycling with in situ x-ray absorption or diffraction. (To be clear, I am not suggesting/requiring new experiments as a condition of publication.)

-page 12, line 257 - The lower-limit TOF value is calculated considering all the metal atoms, but the authors discuss that the current consensus is that Ni does not contribute to the activity (i.e. Ni is not an active OER site). Why not consider Co+Fe atoms only?

-page 13, line 261 - "significantly lower" TOF values - please change to something other than "significantly" unless you are referring here to the statistical definition.

-page 13, line 267 - mass activity given in A/mg of catalyst - how was the mass loading determined? Is this from an estimate/calculation or from QCM measurements? If QCM, then this would also include mass from intercalated water molecules and borate anions (and Ni atoms), which are not directly involved in the OER process. It might be worthwhile to comment on the meaningfulness of conventional mass activity values for these types of catalysts compared to precious metals.

-page 14, line 282 - "extremely harsh OER environment" - how are these conditions relevant for applications? You mention this a bit later in the conclusions, but it would be good to explain here why these conditions in particular were tested (e.g. explain the significance/practicality for CO₂RR - this is not necessarily obvious to the reader)

-page 14, line 304 - "2.01% to 3.04%" - is the hundredths of a percent digit significant? How do these values compare to the current state-of-the-art?

-page 17, line 368 - there is a typo "apromimately" instead of approximately

-page 18, line 393-394 - there are two typos - "giving time" should be changed to "given time"

-page 20, lines 417-424. The description of analytical methods is lacking in detail. Please expand (this can be done in the Supporting Information file). For example, with XPS: What was the x-ray anode source used? What were the data collection parameters (e.g. step size) and analysis methods used for fitting? How was the binding energy scale calibrated?

-page 23, line 502 - Reference 33 is cited in the caption of Figure 3f comparing mass-activity to "literature-reported values", but I suspect this might be the wrong citation, as it is a very old article looking at one material and thus would not have many reported values of different materials. Please check that this reference and the others throughout the manuscript are correctly indicated and that

references did not get accidentally get shuffled around.

-page 26, line 552 - "...for the failure on self-healing..." - should "on" be changed to "of"?

General comments/suggestions regarding the figures:

-The different line colors are not easily distinguishable if printed in black-and-white/greyscale. Use of different line styles may be helpful for some figures with multiple lines (e.g. dashed, dotted, and solid lines)

-Some figures (e.g. Figure 1c) use red and green coloring together, which will not be distinguishable for readers with common color-vision deficiencies.

-Some legends (e.g. Figure 2b, Figure 3c) are in reverse order top-to-bottom as how the different values appear in the figure. It could be more intuitive if the legend order matches the data order.

-Figure 3d and 3e have the same legends but use different symbols for the same experiment (e.g. KOH is squares in 3d but circles in 3e). It would be more intuitive to be internally consistent with both the shapes *and* the colors.

-Figure 3f - please make either the non-precious or precious data points a different symbol (e.g. open circle) to make it easier to see which is which.

In the supporting information file:

- "ml" is used frequently but should it rather be "mL" for milliliter?

- Please see general comments on the figures as described above. The same general comments regarding color use and legends apply to many of the figures in the SI as well. Distinguishing the different curves is difficult in greyscale.

- page 14, Fig S7 - Did you happen to also measure the C 1s, N 1s and S 2p spectra? It would be interesting to see if the as-deposited samples showed evidence for carbonate/nitrate/sulfate in the interlayer, and then if these was replaced by borate after the electrochemical cycling. (Additional experimentation is not required for publication.)

-page 23, line 293 - there is a typo - "out proposed" should be "our proposed"

-page 25, Figure S17 - "Transmittance" is misspelled on the y axis

-page 28, line 328 - typo - "elements" should be singular

Response to the reviewers' comments

Manuscript ID: NCOMMS-21-21000

Manuscript type: Research Article

Title: "A self-healing catalyst for electrocatalytic and photoelectrochemical oxygen evolution in highly alkaline conditions"

Correspondence Authors: Ian D. Sharp, Yanbo Li

Authors: Chao Feng, Faze Wang, Zhi Liu, Mamiko Nakabayashi, Yequan Xiao, Qiugui Zeng, Jie Fu, Qianbao Wu, Chunhua Cui, Yifan Han, Naoya Shibata

[The reviewer comments are shown in *italic*; responses are in blue; all revisions in the MS and SI are highlighted in red]

Reviewer #1

Comment 1: *The self-healing concept of transition metal OER electrocatalysts is an important area of study, however the underlining premise of this manuscript has been introduced by Chung et al. (ref 23). In this past work, iron (II) ions are added into the OER electrolyte to replenish iron into iron-M hydr(oxy)oxide electrocatalysts, where M includes both Ni and Co. This manuscript does describes this past work with NiFe, but does not refer to the improvement found with FeCo, which is particularly relevant to the authors' ternary FeNiCo-hydr(oxy)oxide electrocatalyst self-healing with similar iron(II) in the OER electrolyte. While the ternary shows even better improvement, this reviewer found the work lacking sufficient novelty for Nature Communications.*

Response 1: We appreciate that the reviewer finds the self-healing concept an important area of study for transition metal OER electrocatalysts. We argue that although the work by Chung et al. (*Nature Energy* 5, 222–230, 2020) addresses a similar topic, it does not compromise the novelty of our work due to the following reasons:

- 1) This paper by Chung et al. demonstrated the dynamic stability of Fe catalytic sites in Fe-M (M = Ni, Co, Mn, Cu) hydro(oxy)oxides for OER by adding Fe(III) ions (instead of Fe(II) as used here) into KOH electrolyte. The redeposition of Fe was thought to be achieved by the adsorption of Fe on M hydr(oxy)oxide and the Fe-M adsorption energy was proposed as a unified descriptor for the dynamic stability. This dynamic stability mechanism indeed works for FeNi hydro(oxy)oxide catalyst for the short period of time tested (1 h). However, our results show that the adsorption mechanism is not enough to maintain the stability of the FeNi hydro(oxy)oxide catalyst during long-term operation because of decreasing Fe concentration in the KOH electrolyte due to the precipitation of Fe ions as insoluble Fe oxyhydroxides. Understanding this temporal change of Fe ion concentration is essential to the design of an effective self-healing strategy for long-term operation. Therefore, a more efficient Fe redeposition mechanism is needed, especially when the concentration of Fe ions in the electrolyte becomes extremely low during long-term operation. By contrast, our work introduces a catalyzed redeposition, rather than a simple adsorption mechanism, to attain true self-healing.
- 2) Although FeCo hydro(oxy)oxide was also studied in their work, the proposed self-healing mechanism of FeCo catalyst was the same as for FeNi catalyst, i.e., the redeposition of Fe

was governed by the adsorption of Fe on Co or Ni hydr(oxy)oxide. The catalytic role of Co on the redeposition of Fe was not revealed in their work. The Co-catalyzed self-healing mechanism we propose and investigate is proven to be effective for long-term (1000 h) stable operation. This is because in our proposed mechanism Co not only acts as a site for adsorption but actually catalyzes the Fe redeposition through the following catalytic reactions: $\text{Co(II)} + \text{h}^+ \rightarrow \text{Co(III)}$; $\text{Co(III)} + \text{Fe(II)} \rightarrow \text{Co(II)} + \text{Fe(III)}$. This catalytic effect of Co is crucial for maintaining sufficient Fe redeposition rate with the decreased Fe concentration during long-term operation.

- 3) It is important to note that the basic mechanisms of stabilization are different between this recent Nature Energy publication and our own manuscript. While that work highlights the importance of Fe instability in this class of catalysts and reveals the role of Fe adsorption energetics, the conceptual advance of our work is to identify a new self-healing mechanism afforded by a rationally designed catalytic redeposition that allows for long-term stabilization. Thus, we believe this conceptually new and important catalyst design principle is of sufficient novelty for Nature Communications, as also suggested by both Reviewer #2 and Reviewer #3.

Comment 2: *Additionally, the study used two different substrates (FTO and gold) for different aging conditions which calls into question the role of the substrate. Also, while the role of boric acid is cited to facilitate the dissolution and re-deposition of the iron, the chemistry is not clear.*

Response 2: The reason for using gold instead of FTO substrate for the stability test at high current densities (100 and 300 mA cm⁻²) is because the series resistance iR loss would be too high if the FTO substrate were used. For a typical FTO electrode (1.1 cm²), in which the series resistance (R) is approximately 5 Ω, the iR losses would be as high as 550 and 1650 mV at current densities of 100 and 300 mA cm⁻², respectively. This makes it impractical to test the FTO electrode at high current densities. Therefore, gold electrodes with much lower R value were used. The fact that self-healing of the NiCoFe-B₃ catalyst is achieved on different substrates demonstrates the robustness and general applicability of the self-healing mechanism.

Regarding the role of boric acid in increasing the solubility of Fe, we have added a reference (Xiong *et al. Chem. Geol.* **493**, 16–23, 2018) in Supplementary Table S1. According to this work, the solubility of Fe species increases in the KOH electrolyte with the presence of borate ions, which is attributed to the formation of an aqueous ferrous iron borate complex [FeB(OH)⁴⁺]. With the increased Fe ion concentration in the electrolyte, the Fe redeposition rate is hence enhanced.

Reviewer #2

General Comment: *Very interesting work about self-healing OER catalyst that is relatively stable (1000h at 10mA-cm⁻²) in alkaline media. Shown results, including approach to catalyst design is of relevance for the field. Work is filling existing gap in the literature, discussing in very relevant manner stability of self-healing catalysts. Results are supporting conclusions and claims. Paper is well written with adequate graphic material. I could recommend acceptance in current form, however it would be better if authors deal with several comments I have included in the report:*

Response: We genuinely appreciate the reviewer for their very positive evaluation of our work and for recommending our work for publication. We further would like to thank the reviewer for

constructive comments which helped to improve the quality of the paper. We have considered all suggestions and have modified the manuscript accordingly, including via the collection and inclusion of additional data. The detailed responses to the specific comments are provided below.

Comment 1: *On page 3 lines 34-36, authors stated: “Although these value-added chemicals are produced from water or CO₂ reduction reactions taking place at the cathodes, water oxidation at anodes is required to provide an abundant source of electrons and protons and complete the overall reaction” This is incorrect sentence. If you address reactions alkaline media (like in the title is mentioned) then HER is based on cathodic water reduction, however, OER is based on anodic hydroxyl ion oxidation, not on anodic water oxidation.... water oxidation proceeds in acidic media..... So, please correct this.*

Response 1: We appreciate the reviewer for pointing out this mistake. We have changed "water oxidation at anodes" into "anodic hydroxyl ion oxidation" on page 3 line 7. Following this comment, we have also revised the term "water oxidation" in the title of the manuscript into "oxygen evolution".

Comment 2: *Again, page 3 lines 40-43, authors talk about water oxidation in alkaline media...Please make this consistent, according to comment 1.*

Response 2: Thank you again for this comment. We have changed "water oxidation" into "OER" on page 3 line 12.

Comment 3: *On page 4 authors assume that Co will alter redox potential of Fe redeposition. How exactly?*

Response 3: As shown in Fig. 2f, the redeposition of Fe occurs when Fe(II) is oxidized into Fe(III)OOH. According to Ref. 33, Co exhibits a catalytic effect on the oxygenation of Fe(II) into Fe(III) through the following reaction: $\text{Fe(II)} + \text{Co(III)} \rightarrow \text{Fe(III)} + \text{Co(II)}$. While Co(II) can be electrochemically oxidized into Co(III) at much lower potential (as shown in Supplementary Fig. S4b): $\text{Co(II)} - e^- \rightarrow \text{Co(III)}$, this lowers the redeposition potential of Fe.

Comment 4: *Page 4, line 71 authors claim extremely harsh alkaline conditions, however, 10mAcM-2 is not even harsh condition..... I would definitely not use formulation “extremely harsh”*

Response 4: According to the reviewer's suggestion, we have removed "extremely harsh" from the sentence.

Comment 5: *Authors claim self-limiting thickness.... This should be at very limited potential range.... As overpotential increases the larger will be potential difference between OER potential and redeposition potential for Fe. Is this wrong assumption or ...?*

Response 5: Yes, the self-limiting thickness is indeed a property at potential range below ~1.7 V vs. RHE. As shown in Fig. 1c, as the potential increases above ~1.7 V vs. RHE, there will be appreciable Fe deposition that could increase the thickness of the catalysts. However, as the goal is to achieve efficient OER at potentials as low as possible, this property is valid at potentials of relevance. Especially for PEC applications, the applied potential should be as low as possible to achieve a high solar-to-fuel conversion efficiency.

Comment 6: Page 5, line 91, authors claim intercalation of borate ions ($B_4O_5(OH)_4^{2-}$)... is this really possible?

Response 6: The intercalation of anions in NiFe-LDHs has been widely reported. For instance, Muller et al. (*Energy Environ. Sci.* **9**, 1734-1743, 2016) synthesized NiFe-LDHs nanosheets with different interlayer anions (SO_4^{2-} , CO_3^{2-} , NO_3^- , PO_4^{3-} , BF_4^- , Cl^- , ClO_4^- , ClO^- , $C_2O_4^{2-}$, F^- , I^-) and studied their role in water oxidation catalysis (Fig. R1A). Zhou et al. (*Nano Res.* **11**, 1358–1368, 2018) synthesized 6 different anion-intercalated NiFe-LDHs by a co-precipitation process (Fig. R1B). Different LDHs with borate intercalation have also been reported, such as Mg_2Al-B_i -LDH (Franco de Castro et al. *New J. Chem.* **44**, 10042-10049, 2020), Zn_2Al-B_i -LDH and Mg_3Al-B_i -LDH (Wang & O'Hare, *Chem. Commun.* **49**, 6301-6303, 2013), and NiFe- B_i -LDH (Su et al. *J. Colloid Interface Sci.* **528**, 36-44, 2018). Especially, the widening of the interlayer spacing that matches the size of the borate ions has been revealed by XRD results (Fig. R1C). Therefore, it should be possible for borate ions to intercalate in the LDH layers.

Fig. R1 | Intercalation of anions in NiFe-LDHs. **A**, Schematic illustration of the NiFe-LDH structure with anions and water presented in the interlayer space (Muller et al. *Energy Environ. Sci.* **9**, 1734-1743, 2016). **B**, Relationship between onset potential measured at 1 mA/cm^2 with the standard redox potential of 16 different intercalated anions (Zhou et al. *Nano Res.* **11**, 1358–1368, 2018). **C**, XRD patterns of (a) Zn_2Al -borate LDH and (b) Mg_3Al -borate LDH, and (c) schematic illustration of the structure of $[B_4O_5(OH)_4]^{2-}$ with the LDH layers (Wang & O'Hare, *Chem. Commun.* **49**, 6301-6303, 2013).

Comment 7: On page 6 authors claim opposite to my assumption in comment 6. From EQCM data no redeposition before 1.62 V was detected.... It seems that directly oxide is redeposited, definitely not metal that gets oxidized....Could authors comment such positive potential of Fe deposition.? How that reaction looks like?

Response 7: Even through the redox potential of the $\text{Fe}^{2+}/\text{Fe}^{3+}$ is low, previous studies have revealed that Fe^{2+} is not oxidized readily on some anode materials without a catalytic effect. The kinetics of electrochemical oxidation of $\text{Fe}^{2+}/\text{Fe}^{3+}$ on various anode materials has been studied by Cifuentes and Glasner (Cifuentes & Glasner, *Rev. Metall.* **39**, 260–267, 2003). As shown in Fig. R2, although oxidation of $\text{Fe}^{2+}/\text{Fe}^{3+}$ occurs readily on anode materials with such a catalytic effect at low applied potentials (e.g., Pt, IrO_2 , RuO_2), the current–potential curve for an anode material without a catalytic effect (Pb) shows no oxidation of Fe^{2+} at potentials below that at which oxygen evolution occurred. It was later revealed by Tjandrawan and Nicol (Tjandrawan & Nicol, *Hydrometallurgy* **131–132**, 81–88, 2013) that the oxidation of $\text{Fe}^{2+}/\text{Fe}^{3+}$ occurred at lower potentials on Pb alloy anode only when PbO_2 or MnO_2 species were formed on the anode (Fig. R3). The PbO_2 and MnO_2 were found to catalyze the oxidation of $\text{Fe}^{2+}/\text{Fe}^{3+}$ through the following reactions:

The reason for the high overpotential for the oxidation of $\text{Fe}^{2+}/\text{Fe}^{3+}$ on an anode that does not exhibit such a catalytic effect is probably due to the low rate constant ($4 \text{ M}^{-1} \text{ s}^{-1}$) for the electron exchange reactions of $\text{Fe}^{2+}/\text{Fe}^{3+}$ complexes, which is because the metal-water bond lengths in hydrated Fe^{2+} are longer than those in Fe^{3+} (an example of electrostriction). In contrast, the rate constant for the redox reaction: $\text{Co}^{3+} + \text{Fe}^{2+} \rightarrow \text{Co}^{2+} + \text{Fe}^{3+}$, is significantly higher ($2 \times 10^{18} \text{ M}^{-1} \text{ s}^{-1}$) than that for the redox reaction of $\text{Fe}^{2+}/\text{Fe}^{3+}$ (Brezonik & Arnold, *Water Chemistry: An Introduction to the Chemistry of Natural and Engineered Aquatic Systems*, Oxford University Press, pp439-441, 2011).

In our EQCM and in-situ sequential deposition experiments, bare Au or FTO electrodes were used. These surfaces are unlikely to have a catalytic effect on the oxidation of $\text{Fe}^{2+}/\text{Fe}^{3+}$. Therefore, a high potential is required for Fe deposition through the following reaction:

Fig. R2 | Current density versus electrode potential curves for $\text{Fe}^{2+}/\text{Fe}^{3+}$ oxidation on various anode materials in 0.5 M H_2SO_4 with 1.0 M FeSO_4 at $T = 50\text{ }^\circ\text{C}$ and a magnetic stirring speed of 710 rpm (Cifuentes & Glasner, *Rev. Metall.* **39**, 260–267, 2003).

Fig. R3 | Voltammetric scans of Pb-Ca-Sn alloy anode in solution of 150 g L^{-1} H_2SO_4 and 5 g L^{-1} iron(II) with various manganese concentrations (Tjandrawan & Nicol, *Hydrometallurgy* **131–132**, 81–88, 2013).

Comment 8: On page 7, line 136, authors accept idea from Ref.33 that some cations (Co, Mn, Cu) catalyze Fe-deposition. They prove it by enhanced deposition of Fe in presence of Co. However, they introduce boron based anion (borate) to increase distance between basal planes.... Is it really fact that borate ions get intercalated as stated in Re.34. Is there any proof for that?

Response 10: We have tried in-situ XRD measurement in order to see if we can observe XRD patterns similar to those shown in Fig. R1C. Unfortunately, due to the thin thickness of our catalyst layer, it was not possible to detect any XRD signal. However, as we argued in Response 6, anion intercalation is a common phenomenon in LDH materials and similar borate-intercalated LDHs have been synthesized. Thus, it is entirely feasible that borate ions are intercalated in the LDH layers, which is consistent with previously synthesized materials as well as the electrochemical observations reported in our work.

Comment 9: Authors used Ag/AgCl reference...this reference electrode should be avoided in alkaline-media.... specially if concentration of base is above 0,1 M. Check for example <https://doi.org/10.1002/cssc.201702287>

Response 9: Indeed, conventionally Hg/HgO reference electrode is more suitable in alkaline-media than the Ag/AgCl reference electrode. We did use a Hg/HgO reference electrode at the beginning of our experiment. However, we found that the potential of the Hg/HgO electrode fluctuated with the variation of room temperature because the potential sensing part of the

Hg/HgO electrode was exposed to air. During the long-term experiments, it was difficult to maintain constant room temperature. A variation of room temperature by 5 °C would cause a potential fluctuation of over 10 mV. As the change of potential due to the degradation of the catalysts was small (as shown in Supplementary Fig. S2b), it would be difficult to judge whether the change of potential is due to the change of room temperature or due to the degradation of the catalysts. With Ag/AgCl reference electrode, the potential sensing element was immersed into the electrolyte, whose temperature was well-controlled at 20 ± 0.1 °C using a constant temperature water circulator. The potential variation due to the change of room temperature was therefore eliminated. We have also checked the potential of the Ag/AgCl electrode before and after the stability tests to make sure the variation was within 3 mV. The Ag/AgCl electrode was periodically replaced during the study. Ideally, Hg/HgO reference electrode with the potential sensing part that can also be immersed into the electrolyte should be used. We will see if it is possible to design such a Hg/HgO reference electrode for future studies.

Comment 10: *Authors did not mention effective surface area due to gas-bubble blockage and how this phenomenon influences their results.*

Response 10: We noticed that the gas-bubble blockage problem was not severe in our case as the generated O₂ bubbles were quickly released from the surface of the electrode, as shown in the video below:

Bubbles.m p4

(double-click to play)

Reviewer #3

General Comment: *This work by C. Feng et al. describes the method for depositing and characterization of a "self-healing" OER catalyst containing Co, Fe, and Ni in a LDH structure with borate ions intercalated in the interlayer spacing. The self-healing properties are imparted by dissolved Fe(II) in the electrolyte solution which undergo catalytic reaction with Co sites in the thin film to redeposit Fe into the catalyst layer. The presence of borate ions in the electrolyte is also important for the catalyst's self-healing ability.*

Overall, I found the paper to be well written, with clear hypothesis-driven experiments leading to conclusions that are well-supported by the data. The methods are sound, and the interpretation of the results is appropriate and conclusions/implications are not overstated or sensationalized.

*The authors have done a nice job of tying together some loose ends in the field to show conclusively that the presence of Co in the LDH catalyst film is necessary to stabilize the dynamic Fe processes and maintain high activity of the materials. However, each of these items individually, (1) that Fe in solution is necessary for sustained performance, and (2) that ternary NiCoFe LDH catalysts are more stable than NiFe or CoFe, are already well known in the field. So while I am struggling to find faults with anything in the paper, I think the authors need to better emphasize what they think are the *particularly noteworthy, new results* here so that it is clear why this work merits publication in Nature Communications.*

For example, the distinction that soluble Fe(II) rather than Fe(III) is necessary seems to be a

new finding, and that including borate ions in the electrolyte is integral for halting performance degradation is also new, yet neither of these items are mentioned in the abstract. I think the authors need to clearly highlight what sets this work apart so that it doesn't read like just another iteration of making a new catalyst formulation and testing it in a PEC device, of which there are so many papers coming out every week in this very crowded field. I think there is a lot of novelty and important findings in here already, it is just a bit buried.

Otherwise, I have a very few minor comments/suggestions which are outlined below.

Response: We greatly appreciate that the reviewer finds our paper well-written and conclusions well-supported by the data. We further would like to thank the reviewer for the suggestion to better emphasize the novelty and important findings of our work. In the revised manuscript, we have highlighted the roles of Fe(II) and borate ions in the abstract. The minor questions raised by the reviewer have also been fully addressed, as described below. We sincerely hope that the details provided below will address the reviewer's concerns.

Comment 1: - page 8, line 148 - What is the mechanism of electrodeposition of the NiCoFe-Bi LDHs under anodic conditions? Is it known? Can the authors provide a reference or reasonable hypothesis?

Response 1: The oxidative electrodeposition of thin-film transition metal (oxy)hydroxides has been reported previously (Morales-Guio *et al. J. Am. Chem. Soc.* **138**, 8946-8957, 2016). We have added this reference as Ref. 35 in the revised manuscript.

Comment 2: - page 8, line 150 - The films are described as being "a few nanometers" thick. Can you be more specific with the film thickness, and how was the thickness determined? Were any substrate-induced effects observed for thin films? (i.e. was OER activity influenced by substrate for films of a given thickness?)

Response 2: We thank the reviewer for this important comment. Previously the thickness of the catalyst layer was estimated from SEM observation. The estimated thickness was not accurate due to the limitation of the resolution. Following the reviewer's comments, we have employed STEM and HRTEM to determine more accurately the thickness of the catalyst layer, which is approximately 35 nm. The results are provided in Supplementary Fig. S3.

Supplementary Fig. S3 | Structural properties of NiCoFe-B_i catalyst electrodeposited on FTO substrate. **a**, STEM image of a NiCoFe-B_i layer on FTO substrate and corresponding energy-dispersive X-ray spectroscopy (EDS) mappings of **b**, Ni, **c**, Co, and **d**, Fe. **e**, HRTEM image of the NiCoFe-B_i layer on FTO substrate. The thickness of the NiCoFe-B_i layer is approximately 35 nm.

To determine if there are any substrate-induced effects, we have compared the OER activity of NiCoFe-B_i catalysts deposited on Au and FTO substrates. As shown in Fig. R4, although the apparent OER activity is much better for the sample deposited on Au electrode, the polarization curves are quite similar after iR correction, suggesting the intrinsic OER activities are similar for catalysts deposited on Au and FTO. Therefore, there is not an obvious substrate-induced effect, at least for Au and FTO substrates.

Fig. R4 | The OER polarization curves of NiCoFe-B_i catalyst deposited on Au and FTO substrates: a, without iR correction; b, with iR correction.

Comment 3: - page 11, line 220-222 - "The drastic structural rearrangement...." Have you measured the basal spacing (i.e. interlayer distance) in the NiCoFe-Bi catalyst? Or could you estimate it based on the size of the borate anion as depicted in your figures? It looks quite bulky compared to e.g. chloride or nitrate or sulfate, which are often the interlayer anions present during synthesis of these types of LDHs. I am wondering if the borate itself is so large that it prevents collapse/expansion of the structure during cycling and this is its contribution to enhancing the catalyst stability. One could attempt to synthesize the LDH with some other large anion in the interlayer to see if a similar effect is observed. Or perhaps monitor changes in the structure during cycling with in situ x-ray absorption or diffraction. (To be clear, I am not suggesting/requiring new experiments as a condition of publication.)

Response 1: We appreciate the reviewer's suggestion for this interesting experimental direction. Unfortunately, due to the thin thickness and the low crystallinity (as can be seen from the HRTEM image in Supplementary Fig. S3e) of the electrodeposited NiCoFe-B_i catalyst layer, it was not possible to detect any XRD signal from the sample. Risch et al (*Chem. Commun.* **47**, 11912–11914, 2011) have also reported that for electrodeposited Ni-B_i catalyst, it was not possible to

detect the interlayer spacing by XRD, presumably due to irregular layer spacing caused by the presence of relatively high amounts of borate. However, their results suggested it might be possible to reveal the interlayer spacing by studying the Ni coordination environment by x-ray absorption spectroscopy (XAS). Unfortunately, XAS is not readily available to us within a short time span for revising the manuscript. However, we are planning to carry out in-situ XAS, EXAFS, and GIWAXS experiments in the next available cycle to see if it possible to monitor the structural changes and to study the effects of other large anions, as suggested by the reviewer.

Comment 4: - page 12, line 257 - *The lower-limit TOF value is calculated considering all the metal atoms, but the authors discuss that the current consensus is that Ni does not contribute to the activity (i.e. Ni is not an active OER site). Why not consider Co+Fe atoms only?*

Response 4: Although we consider that Ni is not an active site, the Ni(III)/(II) redox couple contributes to the OER activity by providing a conduit for transporting electrons to the electrode (Hunter *et al. Joule*, **2**, 747-763, 2018). As there are also on-going debates about the role of Ni, in order not to overestimate the TOF, we calculated the lower-limit value by counting all the transition metal atoms.

Comment 5: - page 13, line 261 - *"significantly lower" TOF values - please change to something other than "significantly" unless you are referring here to the statistical definition.*

Response 5: We thank the reviewer for this comment. In the revised manuscript we have provided the exact TOF values (0.14 and 0.22 s⁻¹) instead of using the word "significantly".

Comment 6: - page 13, line 267 - *mass activity given in A/mg of catalyst - how was the mass loading determined? Is this from an estimate/calculation or from QCM measurements? If QCM, then this would also include mass from intercalated water molecules and borate anions (and Ni atoms), which are not directly involved in the OER process. It might be worthwhile to comment on the meaningfulness of conventional mass activity values for these types of catalysts compared to precious metals.*

Response 6: The mass loading was determined by ICP-MS results. As the molar area densities of Fe, Co, and Ni were measured in Fig. 3c, the mass of the metal atoms on the electrode can be calculated.

Comment 7: - page 14, line 282 - *"extremely harsh OER environment" - how are these conditions relevant for applications? You mention this a bit later in the conclusions, but it would be good to explain here why these conditions in particular were tested (e.g. explain the significance/practicality for CO₂RR - this is not necessarily obvious to the reader)*

Response 7: We thank the reviewer for this important comment. We have explained in more detail here the relevance of test conditions for applications of alkaline water electrolysis and CO₂RR electrolysis:

“The maximum concentration of KOH tested here (9.5 M, ~39 wt.%) is higher than those used in commercial alkaline water electrolyser (5.4-8.2 M, 25-35 wt.%)⁴³. In state-of-the-art CO₂RR catalysis systems, highly concentrated KOH electrolyte (1-10 M, 5.3-40 wt.%) was also commonly used to fine-tune the local reaction environment for an improved C-C coupling⁴⁴⁻⁴⁶. Our results demonstrate that the NiCoFe-B_i catalyst is robust against the harsh environments due to its extraordinary self-healing ability and may find practical applications in alkaline water

electrolysis and CO₂RR electrolysis.”

Comment 8: - page 14, line 304 - "2.01% to 3.04%" - is the hundredths of a percent digit significant? How do these values compare to the current state-of-the-art?

Response 8: We have changed the ABPE values into “2.0% to 3.0% to keep a tenths of a percent digit significant. We have added Supplementary Fig. S20 to summarize the performance (both efficiency and stability) of the current state-of-the-art n-Si photoanodes.

Supplementary Fig. S20 | Efficiency and stability of n-Si photoanodes. The red line indicates a decreasing ABPE during the stability test, while the green line indicates an increasing ABPE during the stability test.

Comment 9: - page 17, line 368 - there is a typo "apromimately" instead of approximately

Response 9: We thank the reviewer for pointing out this typo, it has been corrected accordingly.

Comment 10: - page 18, line 393-394 - there are two typos - "giving time" should be changed to "given time"

Response 10: We thank the reviewer for pointing out these typos, they have been corrected accordingly.

Comment 11: - page 20, lines 417-424. The description of analytical methods is lacking in detail. Please expand (this can be done in the Supporting Information file). For example, with XPS: What was the x-ray anode source used? What were the data collection parameters (e.g. step size) and analysis methods used for fitting? How was the binding energy scale calibrated?

Response 11: We thank the reviewer for this comment. We have provided the details about the XPS in the Methods:

“XPS spectra were taken using a Thermo Scientific ESCALAB 250 Xi with a monochromatic Al K α X-ray source with a beam size of 400 μ m. The core-level spectra were collected with a constant analyzer pass energy of 50 eV and a step size of 0.05 eV and the binding energy was calibrated by setting the binding energy of the hydrocarbon C 1s feature to 284.8 eV. Spectrum analysis was performed with the Thermo Scientific Avantage software.”

Comment 12: - page 23, line 502 - Reference 33 is cited in the caption of Figure 3f comparing mass-activity to "literature-reported values", but I suspect this might be the wrong citation, as it is a very old article looking at one material and thus would not have many reported values of different materials. Please check that this reference and the others throughout the manuscript are correctly indicated and that references did not get accidentally get shuffled around.

Response 12: We thank the reviewer for pointing out this mistake, it has been corrected accordingly. We have also checked the other references thoroughly to make sure they are correctly cited.

Comment 13: - page 26, line 552 - "...for the failure on self-healing..." - should "on" be changed to "of"?

Response 13: We thank the reviewer for pointing out this mistake, it has been corrected accordingly.

General comments/suggestions regarding the figures:

Comment 14: The different line colors are not easily distinguishable if printed in black-and-white/greyscale. Use of different line styles may be helpful for some figures with multiple lines (e.g. dashed, dotted, and solid lines)

Response 14: We thank the reviewer for the suggestion. For lines that are not easily distinguishable in greyscale, we have replaced some of the solid lines with dashed lines (Figs. 1b, 3a, 4d-f, 5c-d, Supplementary Figs. S2a, S3b, S13b, S14b, S15b-c, S16b).

Comment 15: Some figures (e.g. Figure 1c) use red and green coloring together, which will not be distinguishable for readers with common color-vision deficiencies.

Response 15: We thank the reviewer for the suggestion. The figures with both red and green coloring have been redrawn (Figs. 1c, 2b-e, 3a-b, Supplementary Figs. S2, S4, S5, S10, S15).

Comment 16: Some legends (e.g. Figure 2b, Figure 3c) are in reverse order top-to-bottom as how the different values appear in the figure. It could be more intuitive if the legend order matches the data order.

Response 16: We thank the reviewer for the suggestion. The legend order has been rearranged to match the data order (Figs. 3c-d, 5c-d, Supplementary Figs. S4b-d, S7, S8, S10, S11). For the legends in Fig. 2b and Supplementary Fig. S5, since they also indicate the sequence for adding the ions, they are kept unchanged.

Comment 17: Figure 3d and 3e have the same legends but use different symbols for the same experiment (e.g. KOH is squares in 3d but circles in 3e). It would be more intuitive to be internally consistent with both the shapes *and* the colors.

Response 17: We thank the reviewer for the suggestion. The symbols in Fig. 3e have been changed to match those in Fig. 3d.

Comment 18: Figure 3f - please make either the non-precious or precious data points a different symbol (e.g. open circle) to make it easier to see which is which.

Response 18: We thank the reviewer for the suggestion. The precious data points have been changed into open circles in Fig. 3f.

General comments/suggestions in the supporting information file

Comment 19: "ml" is used frequently but should it rather be "mL" for milliliter?

Response 19: We thank the reviewer for pointing out this mistake, it has been corrected in Supplementary Fig. S19.

Comment 20: *Please see general comments on the figures as described above. The same general comments regarding color use and legends apply to many of the figures in the SI as well. Distinguishing the different curves is difficult in greyscale.*

Response 20: We thank the reviewer for the suggestion. The figures in the SI have also been revised accordingly.

Comment 21: *- page 14, Fig S7- Did you happen to also measure the C 1s, N 1s and S 2p spectra? It would be interesting to see if the as-deposited samples showed evidence for carbonate/nitrate/sulfate in the interlayer, and then if these was replaced by borate after the electrochemical cycling. (Additional experimentation is not required for publication.)*

Response 21: We have measured the C 1s spectrum in order to calibrate the binding energy. However, N 1s and S 2p spectra were not measured. We will investigate this effect in future studies.

Comment 22: *- page 23, line 293 - there is a typo - "out proposed" should be "our proposed"*

Response 22: We thank the reviewer for pointing out this typo, it has been corrected accordingly.

Comment 23: *- page 25, Figure S17 - "Transmittance" is misspelled on the y axis*

Response 23: We thank the reviewer for pointing out this typo, it has been corrected in Supplementary Fig. S18.

Comment 24: *- page 28, line 328 - typo - "elements" should be singular*

Response 24: We thank the reviewer for pointing out this mistake, it has been corrected accordingly.

REVIEWERS' COMMENTS

Reviewer #2 (Remarks to the Author):

After careful reading of the revised version of the manuscript by Yanbo Li et al (A self-healing catalyst for electrocatalytic and photoelectrochemical oxygen evolution in highly alkaline conditions), I could recommend publication of this interesting work in Nature Communications. I believe it will be read with great interest.

Reviewer #3 (Remarks to the Author):

The authors have addressed all of my concerns from the original review. I recommend publication of the manuscript in its current form.